# In silico immune infiltration profiling combined with functional enrichment analysis reveals a potential role for naïve B cells as a trigger for severe immune responses in the lungs of COVID-19 patients

Yi-Ying Wu[1], Sheng-Huei Wang[2,3], Chih-Hsien Wu[4], Li-Chen Yen[5], Hsing-Fan Lai[4,6], Ching-Liang Ho[1], Yi-Lin Chiu[4]*

1 Division of Hematology and Oncology Medicine, Department of Internal Medicine, Tri-Service General Hospital, National Defense Medical Center, Taipei, Taiwan (R.O.C.), 2 Division of Pulmonary and Critical Care Medicine, Department of Internal Medicine, Tri-Service General Hospital, National Defense Medical Center, Taipei, Taiwan (R.O.C.), 3 Graduate Institute of Medical Sciences, National Defense Medical Center, Taipei, Taiwan (R.O.C.), 4 Department of Biochemistry, National Defense Medical Center, Taipei, Taiwan (R. O.C.), 5 Department of Microbiology and Immunology, National Defense Medical Center, Taipei, Taiwan (R. O.C.), 6 Graduate Institute of Life Sciences, National Defense Medical Center, Taipei, Taiwan (R.O.C.)

* yc566@georgetown.edu

**Data Availability Statement:** All gene expression profiling files are available from the NCBI GEO

## Abstract

COVID-19, caused by SARS-CoV-2, has rapidly spread to more than 160 countries worldwide since 2020. Despite tremendous efforts and resources spent worldwide trying to explore antiviral drugs, there is still no effective clinical treatment for COVID-19 to date. Approximately 15% of COVID-19 cases progress to pneumonia, and patients with severe pneumonia may die from acute respiratory distress syndrome (ARDS). It is believed that pulmonary fibrosis from SARS-CoV-2 infection further leads to ARDS, often resulting in irreversible impairment of lung function. If the mechanisms by which SARS-CoV-2 infection primarily causes an immune response or immune cell infiltration can be identified, it may be possible to mitigate excessive immune responses by modulating the infiltration and activation of specific targets, thereby reducing or preventing severe lung damage. However, the extent to which immune cell subsets are significantly altered in the lung tissues of COVID-19 patients remains to be elucidated.

This study applied the CIBERSORT-X method to comprehensively evaluate the transcriptional estimated immune infiltration landscape in the lung tissues of COVID-19 patients and further compare it with the lung tissues of patients with idiopathic pulmonary fibrosis (IPF). We found a variety of immune cell subtypes in the COVID-19 group, especially naïve B cells were highly infiltrated. Comparison of functional transcriptomic analyses revealed that non-differentiated naïve B cells may be the main cause of the over-active humoral immune response. Using several publicly available single-cell RNA sequencing data to validate the genetic differences in B-cell populations, it was found that the B-cells collected from COVID-19 patients were inclined towards naïve B-cells, whereas those collected from IPF patients were inclined towards memory B-cells. Further differentiation of B cells between

database or website of the original articles published by Chua et al. (GEO accession numbers: GSE53845, GSE128033, GSE124685, GSE145926, and GSE150316.).

**Funding:** This work was supported by grants from Ministry of National Defense-Medical Affairs Bureau [MND-MAB-109-021 to YLC], Tri-Service General Hospital [TSGH-C108-044 and TSGH-E-109213 to YYW] and the Ministry of Science and Technology, Taiwan (R.O.C.) [MOST108-2311-B-016-001- and MOST109-2320-B-016-004- to YLC; MOST108-2635-B-016-001- to YYW; MOST108-2314-B-016-036- and MOST109-2314-B-016-050- to CLH].

**Competing interests:** The authors have declared that no competing interests exist.

COVID-19 mild and severe patients showed that B cells from severe patients tended to be antibody-secreting cells, and gene expression showed that B cells from severe patients were similar to DN2 B cells that trigger extrafollicular response. Moreover, a higher percentage of B-cell infiltration seems associated with poorer clinical outcome. Finally, a comparison of several specific COVID-19 cases treated with targeted B-cell therapy suggests that appropriate suppression of naïve B cells might potentially be a novel strategy to alleviate the severe symptoms of COVID-19.

## Introduction

COVID-19 (Coronavirus Disease-2019) caused by the SARS-CoV-2 (Severe Acute Respiratory Syndrome coronavirus-2) has rapidly spread to over 160 countries worldwide since the beginning of 2020 [1–3]. Nearly 40 million patients with COVID-19 and over 1,000,000 deaths have occurred up to now. At the same time, the rate of increase in the number of COVID-19 cases is gradually accelerating. Despite the tremendous efforts and resources spent by scientists and clinicians around the world trying to produce vaccines and explore antiviral drugs [4, 5], there is still no potent drug or effective clinical treatment for COVID-19 to date [6]. The clinical indications of COVID-19 disease are quite diverse, symptoms including fever, dry cough, loss of smell or taste, fatigue, diarrhea, conjunctivitis, and pneumonia have been reported [7–9]. About 80% of COVID-19 cases are asymptomatic or exhibit mild to moderate symptoms, about 15% of COVID-19 cases progress to pneumonia [10], and patients with severe pneumonia may die from acute respiratory distress syndrome (ARDS) or multi-organ failure [11, 12]. Moreover, further pulmonary fibrosis caused by SARS-CoV-2 infection-induced ARDS often result in irreversible impairment of lung function, which may leave permanent or semi-permanent damage even if patients recover from COVID-19 [13, 14].

The mechanism by which SARS-CoV-2 causes lung injury is only partially understood. It is believed that SARS-CoV-2 infection activates both innate and adaptive immune responses against the virus. However, excessive inflammatory innate responses and dysregulated adaptive host immune defense may cause harmful tissue damage at sites of SARS-CoV-2 entry, such as the lungs and bronchi, accelerating the process of acute lung injury (ALI) and ARDS. Worse, certain organs, including the lungs, may be irreversibly damaged after SARS-CoV-2 infection [15]. Regarding the association of ARDS with lung fibrosis, ARDS is thought to originate from plasma infiltration into the alveolar lumen due to persistent alveolar epithelial damage [16, 17]. Activation of the coagulation system in the plasma and production of proinflammatory cytokines and chemotactic factors leads to a massive influx of neutrophils, lymphocytes and monocytes/macrophages into the lungs, resulting in the dysregulated release of potent cytotoxic mediators associated with inflammation, which ultimately leads to damage to the lung endothelium and epithelial cells [18]. When epithelial and endothelial cells are damaged, inflammatory mediators are released to further recruit multiple types of inflammatory immune cell infiltrates, prompting fibroblasts to activate and migrate to the wound center while releasing collagen to remodel the extracellular matrix (ECM). Chronic inflammation and persistent repair can trigger an excessive accumulation of ECM components, which will lead to permanent fibrosis. Therefore, it is generally accepted that persistent inflammation due to infiltration of immune cells is a major contributor to pathological fibrosis in the lungs [19].

While maintaining immune defense against SARS-CoV-2 infection in COVID-19 patients is certainly important, the over-activated inflammatory response is also directly linked to poor

prognosis for recovery [20]. If the mechanism by which SARS-CoV-2 infection primarily elicits the immune system response and immune cell infiltration can be found, it would be possible to alleviate the extent of ARDS and lung fibrosis by modulating the infiltration and activation of specific immune cells to attenuate the excessive immune response. We believe that understanding the infiltration status of immune cells in SARS-CoV-2 infected lung tissues is the key to access the originate that raised immune hyperactivation and is also a critical step in developing new therapeutic strategies for COVID-19.

Currently, clinical studies of COVID-19 have focused on the collection and analysis of peripheral blood or bronchoalveolar lavage fluid from patients [21, 22], the extent to which immune cell subsets in the lung tissue of COVID-19 patients are significantly altered remains unclear. The present study applied the CIBERSORT method developed by Newman et al. to comprehensively evaluate the simulated infiltration of 22 immune cells in the lung tissues of COVID-19 deceased patients and to further compare them with immune cells in the lung tissues of patients with idiopathic pulmonary fibrosis (IPF) [23]. By analyzing the immune cell infiltration, we found that multiple immune cell subtypes, especially naïve B cells, were highly infiltrated in the lung tissues of COVID-19 patients. Comparison of functional gene sets revealed that non-differentiated naïve B cells may be the main reason for the overactive humoral immune response. Further analysis of the defined B-cell population using single-cell RNA sequencing databases showed that the B-cells from COVID-19 patients not only tended to be naïve B-cells, but also tended to be antibody-secreting cells in patients with severe disease, and the proportion of B-cell infiltration seemed to correlate with the severity of the disease. We further compared several cases of specific COVID-19 with therapies targeting B cells and found that suppression of naïve B cells is likely to be a new strategy to alleviate the severe symptoms of COVID-19.

## Material and methods

### Evaluation of infiltrating immune cells using CIBERSORT-X

CIBERSORT-X is a powerful tool for simulating tissue immune infiltration in samples using computational algorithms [23–25]. The application of CIBERSORT-X allows accurate quantification of the relative abundance or absolute scores of different immune cell types in complex gene expression mixtures. To characterize and calculate each immune cell subtype, CIBERSORT-X uses approximately 500 genes with consistent gene expression signatures. Here, we applied the original gene signature file LM22, which characterizes 22 immune cell subtypes, including B cells, T cells, natural killer cells, macrophages, dendritic cells, eosinophils, and neutrophils, and analyzed datasets from COVID-19 patient lung tissue and controls from the same database (NCBI GEO accession number: GSE150316), IPF patient lung tissues and healthy donor lung tissues (NCBI GEO accession numbers: GSE53845 and GSE124685) [26, 27]. After normalization of all sample data, the immune cell profiles were analyzed using the CIBERSORT-X web tool to calculate absolute scores, and the mean of each group's immune cell profile was then computed using GraphPad software.

### Identification of B cell specific signatures

For Naive B cell and memory B cell specific signatures, we used LM22 document to define the top 100 enriched genes in subtypes of Naive B cells and Memory B cells respectively, and then analyzed genes in the intersection or non-intersection by Venn diagram. The genes in non-intersection areas were defined as the specific gene set of the cell.

## Single cell RNA sequencing data analysis

For single-cell RNA sequencing data analysis, Bio-turning Browser (BBrowser 2.6.22) software were utilized to download two COVID-19 single-cell sequencing data (GSE145926 published by Liao et al. and another single-cell RNA sequencing data published by Chua et al.) and one IPF single-cell sequencing data (GSE128033 published by Morse et al.) [22, 28, 29]. The annotation of immune cell clusters and disease severity was based on the original author-defined cell clusters and meta-data built in the downloaded data within BBrowser. The matrix of gene expression of mentioned B cells was extracted using BBrowser as well for further GSEA analysis.

## Gene Set Enrichment Analysis (GSEA) and enrichment map visualization

GSEA is a computational method used to investigate whether a given whole gene expression profiling with user defined phenotype is significantly enriched in a set of gene sets [30]. The GSE53845, GSE124685, and GSE150316 databases were downloaded from NCBI and normalized before being input into the GSEA program. The BP:GO biological process (7530 gene sets) and the PID subset of CP (196 gene sets) from the C2CP canonical pathways were downloaded from MsigDB and used as functional gene sets [30]. Samples were categorized into "SARS-CoV-2 vs NegControl" and "IPF vs Healthy donor" according to the original database annotations respectively. $P$-value $<0.05$ and FDR $<0.05$ will be considered statistically significant. Enriched gene sets of GSEA identification on "GO:BP" and "PID pathways" were visualized via the Enrichment Map v3.2.1 plugin in Cytoscape 3.8 [31]. Enrichment maps represent the degree of enrichment of functional gene clusters for each disease group compared to control groups.

For single-cell GSEA analysis, the annotation from the original authors was used to distinguish whether the B cells in matrix were from patients with mild or severe disease. The defined B cell populations were then analyzed using the gene set of "C7:immunologic signature gene sets" describing the B cell lineage in GSEA with default settings. Those that were significantly enriched ($p$-$value < 0.05$, FDR $< 0.05$) in both databases were considered to be the predominant differences in B-cell populations from patients with mild or severe COVID-19 disease. Regarding gene set interpretation, since the C7:immunologic signature gene sets mostly compare the differential gene expression between the two types of immune cells, with "UP" denoting that the moderate is associated with the former and "DN" denoting that the moderate is associated with the latter, we used this principle to define the propensity of B cells in severe disease.

## B cell infiltration and clinical status

For the analysis of B-cell infiltration and clinical status, all data were calculated based on the number of immune cells and clinical status of COVID-19 patients as quoted from S1 Data in the article published by Chua et al.

## Statistical analysis

For CIBERSORT abs scores, the Student's t-test was used for comparisons between the two groups with normally distributed data and the Mann-Whitney test was used for comparisons between the two groups with abnormally distributed data. Pearson correlation analysis was used to estimate the consistency among the 22 immune cell transcriptional estimated infiltration score distributions. Statistical analysis was performed using GraphPad Prism software (GraphPad Software). $P$ $<0.05$ were regarded as statistically significant differences.

## Results

### Rationale and design of an in silico simulated immune cell infiltration profiling study

Nearly all of the patients who died from COVID-19 had severe lung tissue damage and pulmonary fibrosis [32]. On the other hand, mortality in IPF is generally the result of progressive fibrotic lung disease. We believe that comparing the gene expression profiling between lung tissues of COVID-19 and IPF will allow us to access the phenotypes that are specific to SARS-CoV-2 infection. Among these, differences in the level of immune cell infiltration are considered to be the most critical factor in the assessment of an over-active immune system. However, it is difficult to assess experimentally the infiltration of multiple immune cells in clinical. The first step is to obtain ethical approval and valuable COVID-19 lung samples, followed by tissue sampling and analysis in laboratories with adequate biosafety levels, and then staining and analysis of various immune cell populations with specific biomarkers. Moreover, the biomarkers for the analysis were limited, and the proportion of multiple immune cells could not be analyzed simultaneously.

In recent years, algorithms to precisely simulate the proportion of multiple immune cells infiltrating tissue samples using whole gene mapping have emerged and have been used in many studies [33, 34]. The simulation results were confirmed to correlate significantly with the proportion of actual immune cells in several studies [24, 35]. In this study, CIBERSORT was utilized for tissue immune cell infiltration scoring, which was based on 22 types of immune cell subsets profiling, and abs mode was performed to enable cross-database comparison [23]. For sample collection, we used the valuable COVID-19 patient organ RNA-sequencing whole gene expression results uploaded to the NCBI GEO database by Ting et al. as the target for analysis (SARS-CoV-2 infected lung tissue sample N = 16, Negative Control lung tissue sample N = 5). GSEA was then used to evaluate the enrichment score of functional gene sets associated with disease groups. The functional gene set significance filter was set at $P$-value $<0.05$, FDR $<0.05$. Further, Cytoscape Enrichment map was applied to visualize the GSEA results and perform related gene set clustering, the whole process is shown in Fig 1A.

In the lung tissue of COVID-19 patients and negative controls, 22 immune cell infiltration scores were calculated by CIBERSORT and visualized in Morpheus developed by Broad institute [36]. Comparison of the differences by T test showed that T cell CD8+, B cell plasma, monocyte, and Macrophage M1 were significantly increased in the lung tissues of COVID-19 patients (Fig 1B). Similarity matrix analysis of the distribution of 22 immune cells showed that B cell naïve, monocyte, T cell CD8+, B cell plasma, mast cell activated, and T cell CD4+ mem clustered together, indicating that the distribution of these immune cells was similar, and the percentage of infiltration was increased in the lung tissue of COVID-19 patients (Fig 1C).

### Comparison of T-cell lineage transcriptional estimated infiltration in lung tissue between COVID-19 and IPF patients

In order to understand the detailed differences between the immune cell infiltration landscape in COVID-19 and IPF, CIBERSORT-X was carried out on two IPF databases (GSE124685 and GSE53845), and the results were divided into T cell lineage, B cell lineage, Myeloblast lineage, and other cells respectively.

T cell lineage analysis showed significant increases in CD8+, CD4+ naïve, and CD4+ memory activated in both SARS-CoV-2 infection and fibrotic lung tissues (Fig 2). In contrast, there was no consistent comparisons among CD4+ memory resting, regulatory T cell and γδ T cell subgroups.

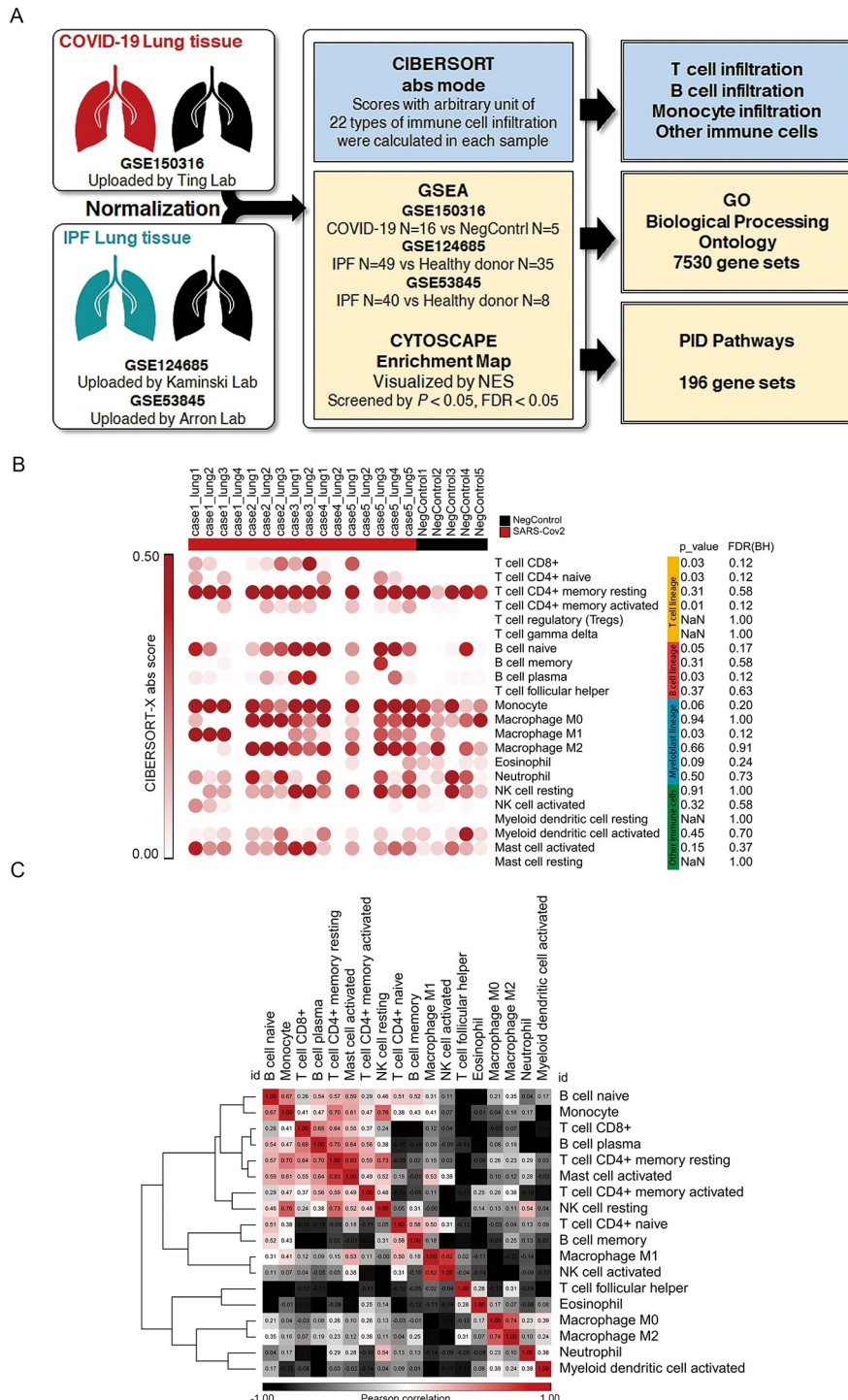

**Fig 1. Overall procedure of the study and transcriptional estimated infiltration of 22 types of immune cells in the lungs of COVID-19 patients.** (A) Diagram showing the comprehensive procedure of the study; (B) CIBERSORT abs score of 22 immune cells, colors represent the classification of the following: T cell lineage, B cell lineage, Myeloblast lineage, and other immune cells; (C) Similarity matrix representing the correlation among 22 immune cells in GSE150316.

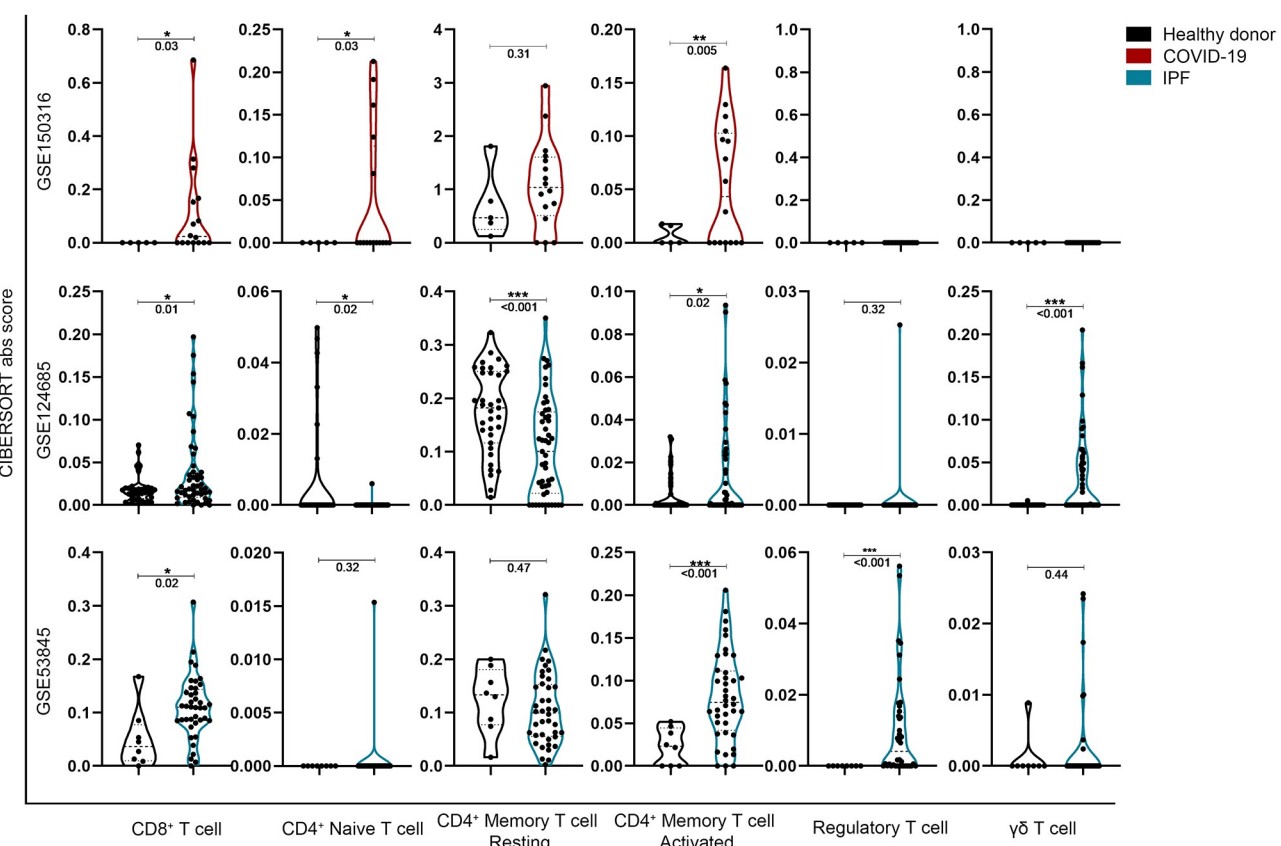

**Fig 2. In silico simulated T-cell lineage infiltration in lung tissue between COVID-19 and IPF patients.** $^*P<0.05$, $^{**}P<0.01$ and $^{***}P<0.001$, with comparisons indicated by brackets.

Related studies showed that CD8$^+$ T cells significantly increased bronchoalveolar lavage fluid in SARS-CoV-2 infected patients [22], and CD4$^+$ memory T cells have been reported to respond to viral spike protein after SARS infection [37]. In addition, both CD4$^+$ naïve T cells and CD8$^+$ T were reported to be significantly increased by mass cytometry (CyTOF) analysis [38].

## Comparison of B-cell lineage transcriptional estimated infiltration in lung tissue between COVID-19 and IPF patients

The results of the B-cell lineage analysis showed that the CIBERSORT abs score of Plasma B cells was significantly increased in both COVID-19 and IPF patients (Fig 3). In contrast, there were significantly elevated naïve B cells in the lung tissue of SARS-CoV-2-infected patients and significantly elevated memory B cells in the lung tissue of IPF patients respectively, suggesting that SARS-CoV-2 infection is associated with the infiltration of naïve B cells rather than memory B cells in the lungs, a phenomenon exactly the opposite of the observations in the IPF. In addition, the infiltration of follicular T cells was significantly increased in one of the IPF databases and was not significantly different in COVID-19 group.

Correlative studies have shown significantly increased expression of memory B cells, plasmablast and BAFF (B cell-activating factor of the TNF family) in lung tissue and peripheral blood of IPF patients [39, 40], where BAFF is considered important for the survival of plasma

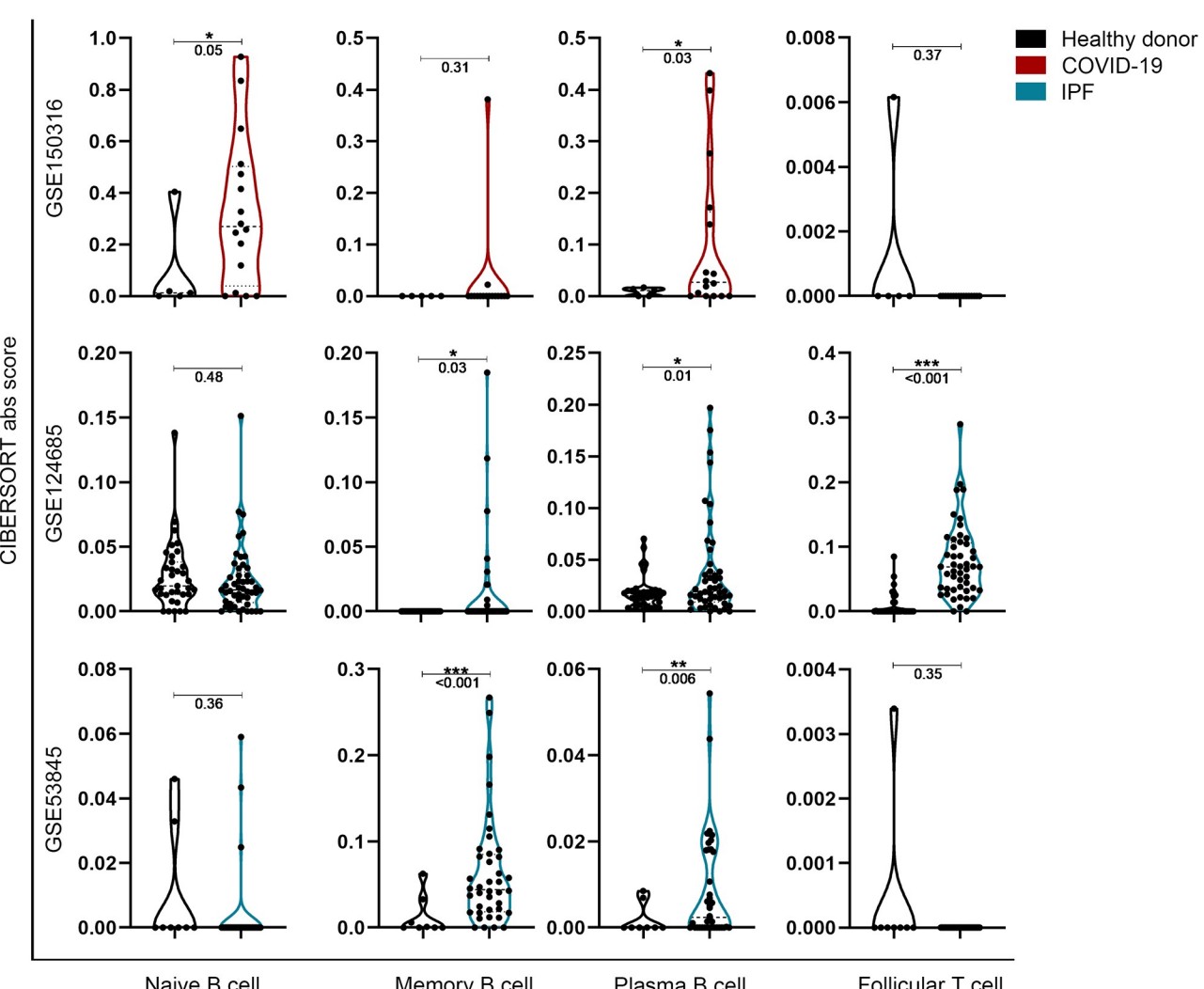

**Fig 3. In silico simulated B-cell lineage infiltration in lung tissue between COVID-19 and IPF patients.** $^*P<0.05$, $^{**}P<0.01$ and $^{***}P<0.001$, with comparisons indicated by brackets.

cells [41]. In addition, the proportion of T follicular helper cells in the peripheral blood of IPF patients increased significantly [42, 43]. In COVID-19 patients, recent studies have shown a significant increase in plasma cells and a significant decrease in naïve B cells in peripheral blood [21]. In an analysis of B cell compartment of SARS-CoV-2 infected patients, Nielsen et al. found that most of the B cells recruited to respiratory tracts in the early stage of infection lacked significant somatic mutation, suggesting that the recruited B cells were similar to naïve B cells [44].

## Comparison of transcriptional estimated myeloblast lineage and other immune cell simulated infiltration in lung tissue between COVID-19 and IPF patients

The analysis of myeloblast lineage showed an increase in monocyte in the lung tissue of SARS-CoV-2 infected patients, and a decrease in the lung tissue of IPF patients (Fig 4). In

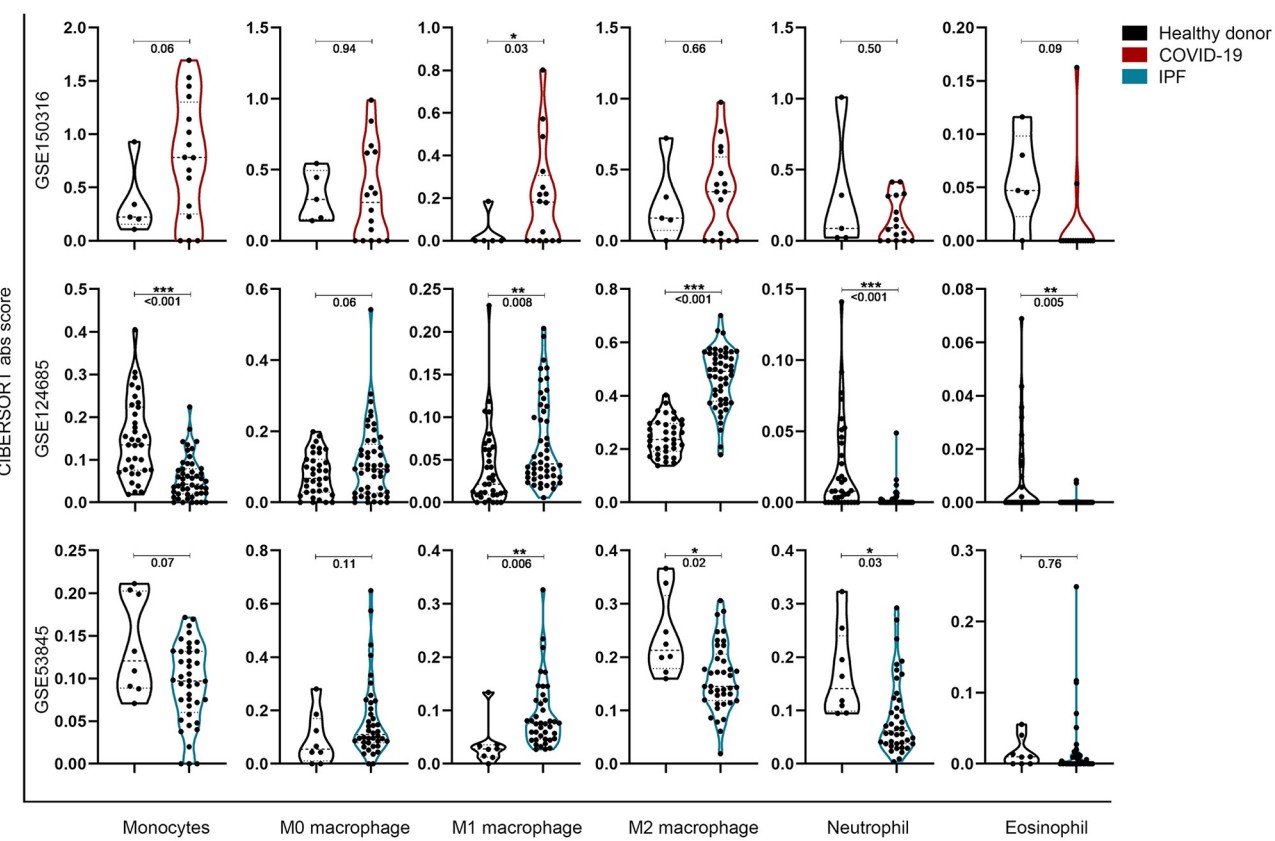

**Fig 4. In silico simulated myeloblast lineage infiltration in lung tissue between COVID-19 and IPF patients.** $^{*}P<0.05$, $^{**}P<0.01$ and $^{***}P<0.001$, with comparisons indicated by brackets.

common, the CIBERSORT-X abs score of M1 macrophage was significantly increased in both disease groups. The results of M0 and M2 macrophage infiltration do not vary significantly or are not consistent between the two IPF databases.

Monocytes and lung macrophages are thought to be involved in the pathogenesis of pulmonary fibrosis [45]. Previous studies have shown that higher monocyte counts observed in the peripheral blood of patients with IPF are significantly associated with poor prognosis [46]. Recent studies have shown that alveolar macrophages (AM), which are primarily involved in the pathogenesis of pulmonary fibrosis, arise more from in situ proliferation than from bone marrow supplementation [47, 48]. These studies suggest that during pulmonary fibrosis, cytokine released from immune cells may locally induce AM polarization to M1 or M2 subtypes and thus influence the progression of fibrosis [49]. The role of monocyte in the lungs of IPF patients remains to be defined.

On the other hand, a high degree of monocyte infiltration has been observed in the lung tissue of COVID-19 patients [22, 38], macrophage infiltration has also been reported recently [50]. The infiltration of monocytes in the lung may be activated by the immune response to form macrophages, which may ultimately promote acute inflammation and cause lung damage through increased M1-polarized macrophages [51]. Other types of immune cells were not significantly different in the lungs of COVID-19 patients, as shown in S1 Fig in S1 Data.

## GO:BP analysis based on GSEA reveals specific enrichment of B-cell-mediated innate humoral responses in the lungs of COVID-19 patients

To understand the similarities and differences in the functional enrichment of gene sets in the lung tissues of COVID-19 and IPF patients compared to controls, the GO:BP (Gene Ontology Biological Processing) gene set was employed to evaluate a variety of biological responses, including immune responses.

Based on the GSEA results, Fig 5A lists the functional gene sets with NES scores greater than 2 (COVID-19: Red, IPF: Green; pre-screening criteria: nom. $P < 0.05$, FDR $< 0.05$). The purple text shows the set of genes that are co-enriched in COVID-19 or IPF patients, including Humoral immune response mediated by circulating immunoglobulin, B cell mediated immunity, complement activation, phagocytosis recognition, positive regulation of B cell activation and B cell receptor signalign pathway ranked first. Showing that an immune response caused by B-cell mediators may be a consistent phenomenon that causes damage to lung tissue in patients with COVID-19 or IPF. Green and red text represent the separately enriched gene sets in IPF or COVID-19 respectively. A number of gene sets related to lung fibrosis, including extracellular structure organization and collagen associated processes, were enriched in IPF. At the same time, a variety of genes related to B-cell proliferation, differentiation and maturation as well as adaptive immune response were also solely associated with IPF. Interestingly, none of these gene sets associated with B-cell maturation and differentiation were prominent in COVID-19, and instead were enriched for the innate immune response and FC receptor signaling pathway. This phenomenon is consistent with the results of the CIBERSORT-X analysis.

To understand which genes affect the CIBERSORT-X algorithm more, we ranked the top 100 genes enriched in naive B cell or memory B cell within LM22 signature document and performed Venn diagram analysis (Fig 5B). There are 20 genes that are not involved in intersection in both types of B cell, we define them separately as "Naïve B cell specific signature" and "Memory B cell specific signature". In addition, to systematically visualize the enriched gene set clusters, we performed the enrichment map app in Cytoscape and set the enriched gene cluster in COVID-19 as red and IPF as green. The visualized results showed that clusters belong to immune cell activation were divergent, while the other clusters were clearly polarized. For example, the chromatin centromere and sister segregation gene sets are enriched only in COVID-19, while others are enriched only in IPF. We further overlap the defined B cell specific signature with the visualized clusters, the graphical representation showed that memory B cell specific signature was mostly associated with multiple immune-related gene sets of IPF. In contrast, the naive B cell specific signature was associated with COVID-19 enriched immune gene sets. These results are consistent with the data from the CIBERSORT-X analysis of B cell lineage (Fig 3). That is, in the lungs of COVID-19 patients, undifferentiated naïve B cells may be the main immune cells that elicit humoral immune response, which has been observed in clinical practice [52, 53].

## PID pathway analysis reveals distinct enrichment between immune cell signaling and migration responses and shared signaling pathways in damaged lung

The function of immune cells is thought to be associated with multiple signaling pathways. Through a comprehensive analysis of the enrichment of signaling pathways, not only the status of specific immune cells can be assessed, but also potential therapeutic targets can be identified. The PID (Pathway Interaction Database) pathway gene set is known for its accuracy in reflecting specific signal pathways, which helps us to precisely define the specific variation in

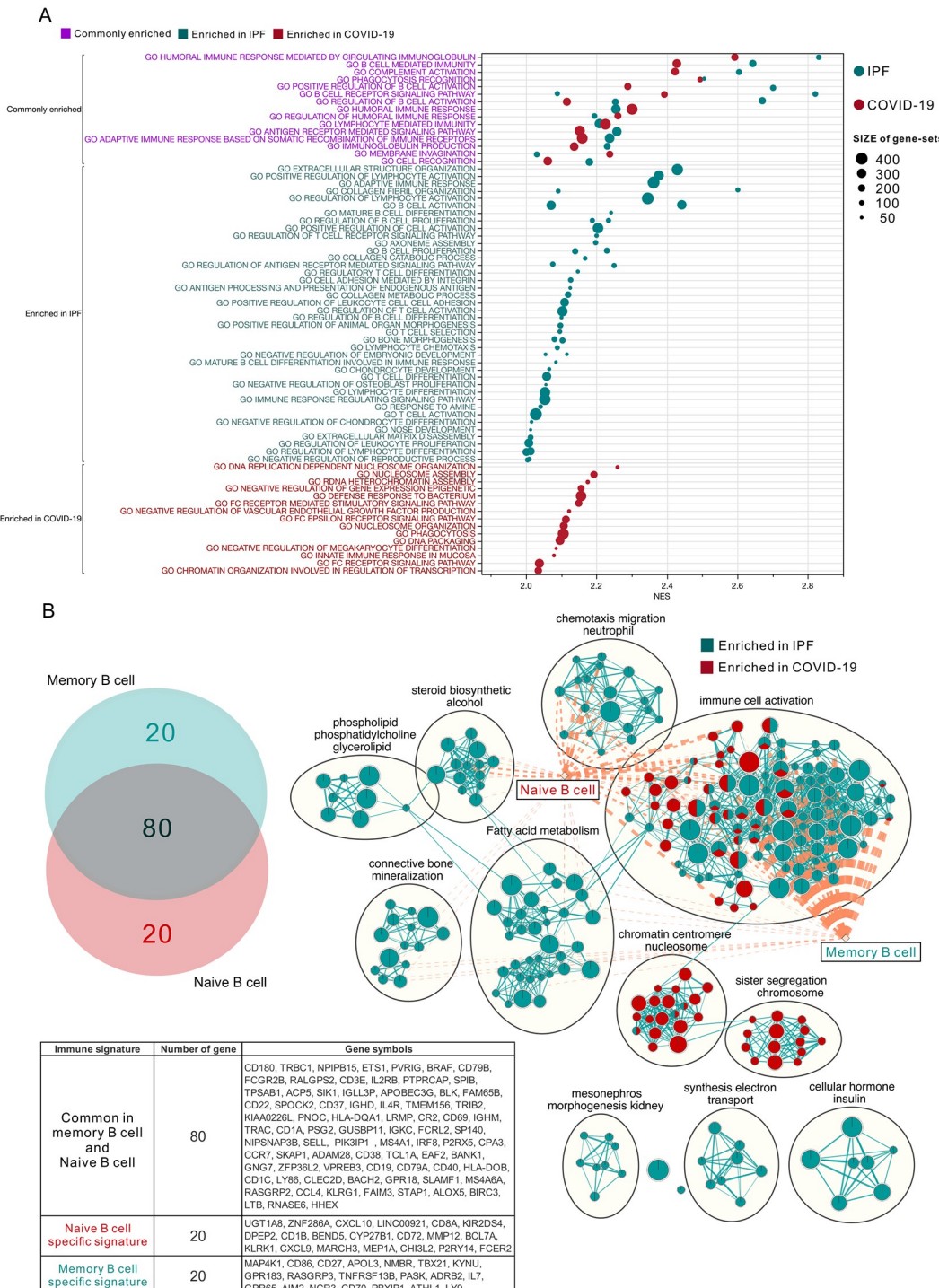

**Fig 5. GO:BP functional enrichment map summary of GSEA in the lung tissues of COVID-19 and IPF patients.** (A) The top NES-ranked enrichment results of biological processing by gene ontology (GO) functional enrichment analysis in IPF or COVID-19 (NES > 2; FDR < 0.05). The normalized enrichment score (NES) was presented in the X-axis, and the names of enriched functional gene sets were shown in the Y-axis. The color gradation on the right side indicates false discovery rate (FDR). Size represents the number of genes in the gene set. Color of the text represents whether the gene set was enriched in both (purple) or in individual type of disease (COVID-19: Red, IPF: Green). (B) Venn diagram: two circles represent the intersection status of the top 100 genes from the Naive B cell (Red) and Memory B cell (Green) subtypes extracted from the LM22 immune cell gene signatures, respectively. The detailed list of genes in the intersection is shown in the table below. Enrichment map: Commonality of positive enrichments in the GO:BP gene sets after GSEA assessment using enrichment

map visualization in COVID-19 (GSE150316, Red) and IPF (GSE124685 and GSE53845, Green). Single color: the node is positively correlated in only one disease condition; Mixed: the node is positively correlated in two or three databases. Diamond: Naïve or Memory B cell specific signature, genes in the signature were shown in the table below. Lines connecting nodes or diamonds represent the degree of overlapping between two gene sets. Criteria for enrichment significance screening: *P*-value <0.05, FDR <0.05.

COVID-19 group [54]. S2 Fig in S1 Data shows the visualized results of the GSEA analysis using PID pathway. In terms of the relevance of immune cell-associated signaling pathways, TCR signaling in Naïve CD4$^+$/CD8$^+$ T cells was significantly activated in both COVID-19 and IPF groups. In addition, CD40/CD40L signaling were negatively correlated only in SARS--CoV-2 infected lung tissues, which was suggested to be associated with the differentiation and proliferation of activated B cells in germinal center [55]. In the cluster of cytokine signaling, IL-1 and IL-6 signaling pathways were negatively correlated, while IL-4 and IL-12 signaling pathways were positively correlated, shows that IL-4 and IL-12 have more significant effects on COVID-19 lung tissue than other cytokines. Among the integrin associated interactions, β1, 3, and 5–7 integrin cell surface interaction were all negatively correlated in COVID-19 group. In contrast, β2 integrin (LFA-1) and α4β1 integrin were positively correlated, which are thought to be essential for B-cell activation and adhesion [56, 57]. Furthermore, among the many signaling pathways, the Insulin pathway showed a significant negative enrichment in both COVID-19 and IPF groups, the physiological significance of which needs to be further verified. Most of the other signaling pathways are unknown or require further study in relation to the immune response, all of which are listed in S3 Fig in S1 Data.

## Differences in B-cell clustering subtypes between COVID-19 and IPF patients verified in single-cell RNA sequencing databases

CIBERSORT and similar deconvolution methods can be used to analyze the distribution of immune cells in the Bulk RNA-seq data, but the analysis of the signal pathways may not be accurate due to the overlap of activation between different cells. In order to further validate the phenomenon observed in bulk RNA-seq, we searched several publicly available single-cell RNA-sequencing databases and restricted the selecting criteria to those sampling from lung-associated body fluids or tissues. Chua et al. published single-cell RNA sequencing analysis of respiratory fluid samples from 20 patients with COVID-19 mild to severe disease, and Liao et al. published single-cell RNA sequencing analysis of bronchoalveolar lavage fluid (BALF) from 9 patients with COVID-19 mild to severe disease (GSE145926). On the other hand, Morse et al. published single-cell RNA sequencing analysis of lung tissues from IPF patients (GSE128033). According to the annotation provided by the authors, we circled B-cell populations from patients or healthy donors and evaluated the overall percentages of expression and the average standardized expression of each gene in the B-cell specific signatures. The average standardized expression of all genes was further analyzed (Fig 6). The results showed that genes in naive B cell specific signature was significantly higher in COVID-19 patient B cells, whereas genes in memory B cell specific signature were generally increased in IPF patient B cells, echoing our observations in bulk RNA-seq databases.

## B cells from patients with severe COVID-19 have genetic phenotype similar to antibody-secreting cells, and the percentage of B cell infiltration appears to associate with the severity of the disease

Given the results of the bulk RNA seq analysis, we were curious whether any specific distinctions existed between the B-cell lineage populations of patients with mild or severe COVID-

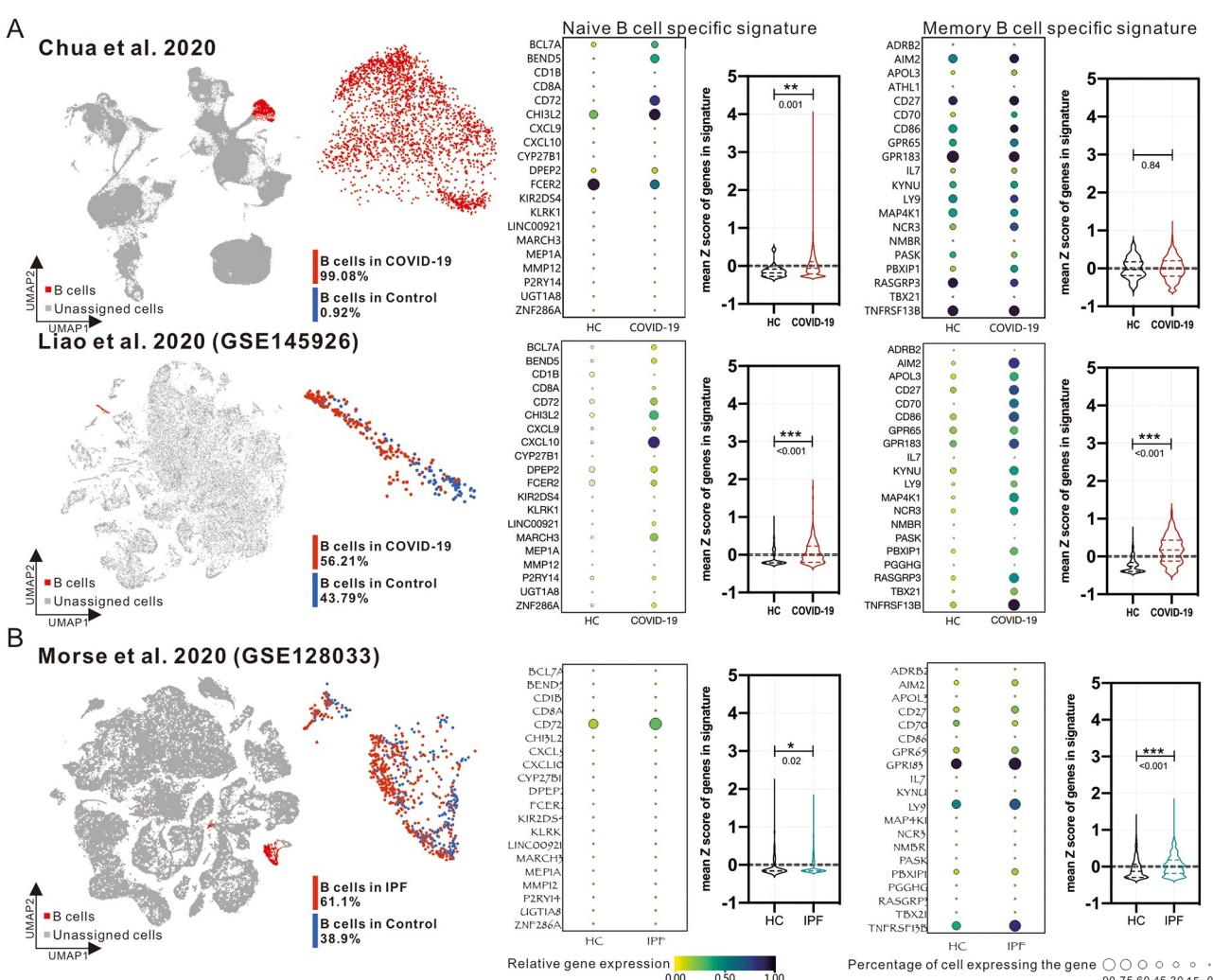

**Fig 6. Validation of the phenotypes of B-cell populations in single-cell RNA sequencing of lung-associated body fluid samples from COVID-19 and IPF patients.** UMAP shows the visualized distribution of whole cells in single-cell RNA sequencing data from two COVID-19 databases (A) and IPF database (B). the "B-cell clusters" annotated by the original authors are shown in red, the unassigned cells are shown in gray. B-cell clusters are further colored red and blue to represent those from disease or healthy donors (HC) respectively. The proportion and average expression of genes related to Naive or Memory B cells in the B-cell population are shown as dot plots. Relative gene expression is shown in color and the overall B-cell expression gene percentage is shown in dot size (%). The mean Z scores of all genes' expression in B cell populations from HC or disease groups are shown in Violin plot. Statistical significance was performed using Welch's t-test approach. *: $P < 0.05$, **: $P < 0.01$, ***: $P < 0.001$. The actual $P$-values are shown under the asterisks as well.

19. Firstly, we divided the B cells from COVID-19 in two single-cell databases by disease severity and analyzed them by GSEA together with gene sets from C7 immune signature that belonged to B cell lineage, and then we cross match the two databases with the gene set enriched in moderate disease. The results showed that B cells from COVID-19 patients with severe symptoms had a tendency to be antibody secreting cells, such as plasma cells, plasmablast, IgM memory B cells, Follicular B cells, and unstimulated B cells compared to those from patients with mild symptoms (Fig 7A). As possessing both naïve B-cell and antibody-secreting B-cell properties, these cells are similar to the DN2 (Double negative) B-cells found in systemic lupus erythematosus proposed by Tipton et al. The researchers found that they are derived from naïve B cells and appear to be precursors of plasma cells [58]. The term double negative

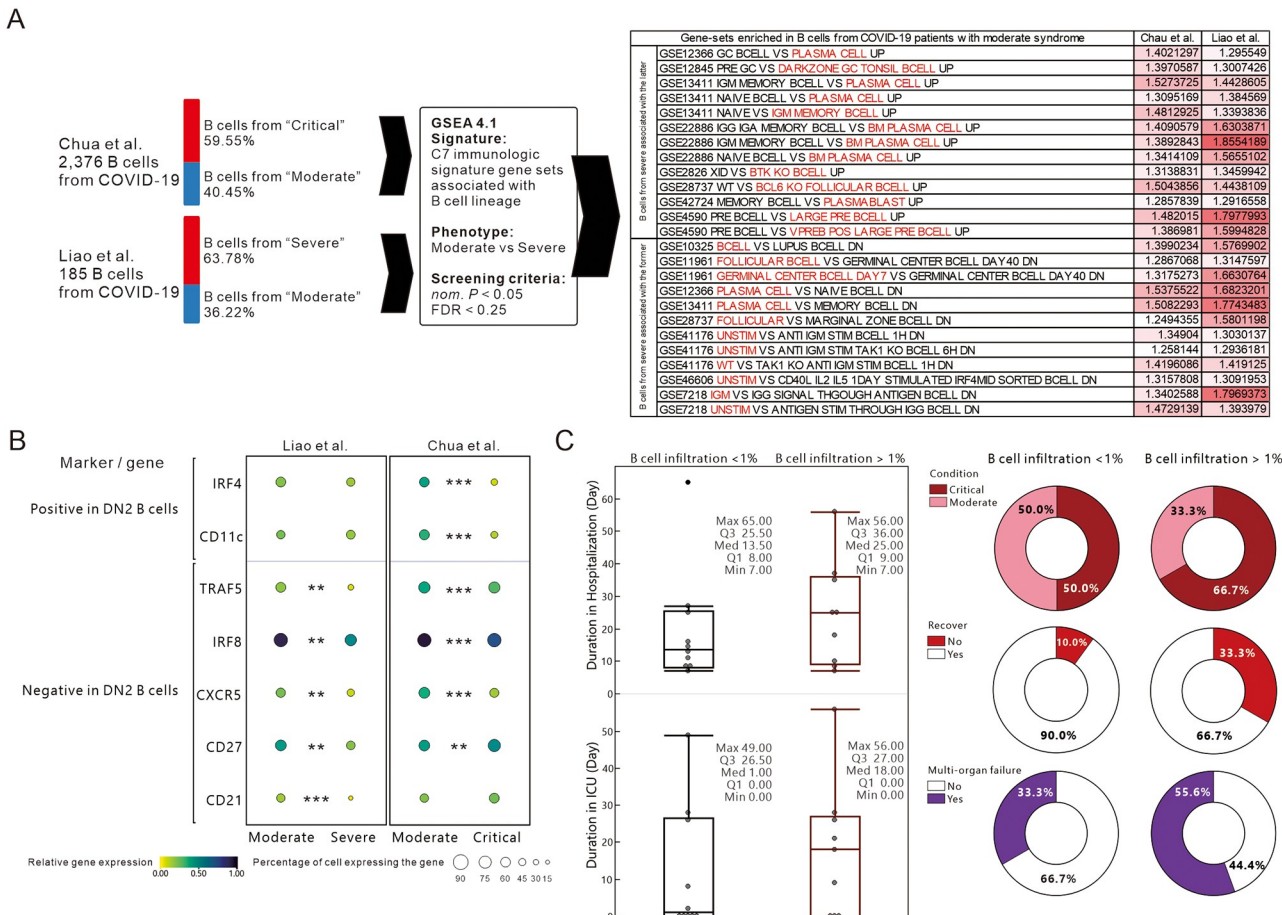

**Fig 7. Assessing the phenotypic and clinical status differences of B-cell populations between mild and severe symptoms of COVID-19.** (A) Schematic diagram of the GSEA analyzing process for B cells of mild and severe COVID-19 patients in different databases. GSEA results were cross-referenced and the overlapping enriched data sets of the two databases were listed. The NES scores of each gene set are additionally represented by shades of red to represent the relative enrichment level. Those considered relevant to the B-cell population of patients with severe COVID-19 in the two immune cell comparisons are marked in red. Criteria for enrichment significance screening: $P$-value <0.05, FDR <0.25. (B) The dot plots represent the overall expression ratio and average expression of genes thought to be related to antibody-secreting cells and DN2 B cells. The overall expression percentage is represented by the size of the dots and the average expression is represented by the color. Welch's t-test was performed to analyze the significance of gene expression in two conditions. **: $P < 0.01$, ***: $P < 0.001$. (C) Comparison of B-cell infiltration ratio and clinical status of patients with COVID-19. Box plots are shown for patients with B cell infiltration ratio < 1% (Black box) or > 1% (Red box) for duration of hospitalization and ICU (Days). Pie charts show the severity, recovery and complications of multi-organ failure in COVID-19 patients with different levels of B-cell infiltration.

(DN) comes from the absence of immunoglobulin D and memory B-cell marker CD27 [59]. DN B cells are further classified into DN1(CXCR5$^+$) and DN2(CXCR5$^-$) by whether CXCR5 is expressed or not [60]. An increase in IRF4 and a decrease in IRF8 are thought to be associated with the promote naïve B cell differentiation into DN2 [61, 62]. The process of induced differentiation is accompanied by a decrease in TRAF5 and CD21 and an increase in CD11c [63]. These DN2 B cells are usually found in B-cell follicles rather than germinal centers and may therefore be genetically phenotypically different from GC-derived B cells. Taken together with the aforementioned observations, we hypothesized that B cells from the lungs of COVID-19 critically ill patients may be associated with DN2 B cells, which have been considered in recent years to be the main inducer of extrafollicular response [63, 64]. In-depth analysis of DN2 B-cell related genes revealed that the overall expression ratio and average expression of genes

that were thought to be reduced in DN2 B cells, such as CD21, CXCR5, IRF8 and TRAF5, were lower in B cells from critically ill patients, especially in samples from BALF (Fig 7B). Further, we analyzed the ratio of B-cell infiltration and the clinical characteristics of the patients and showed that when B-cell infiltration > 1%, there was a longer hospitalization and ICU duration, a higher proportion of severe disease (66.7%), a higher proportion of no recovery (33.3%) and a higher proportion of multi-organ failure (55.6%), indicating an association between increased B-cell infiltration and severity of disease in COVID-19 patients (Fig 7C).

## Discussion

Based on in silico simulation of immune infiltration, this study reveals the impact of SARS-CoV-2 infection on the transcriptional estimated immune infiltration landscape of lung tissue. Significantly increased infiltration of CD4$^+$/CD8$^+$ T cells (Fig 2), Plasma cells (Fig 3) and M1 macrophage (Fig 4) was a common observation in patients with SARS-CoV-2 infection or IPF. It is believed that the increased infiltration of these immune cells is a major contributor to lung damage or fibrosis. However, the elevated naïve B cell infiltration (Fig 3), uncorrelated gene-sets of B cell proliferation and differentiation in the cluster of B cell mediated immune response (Fig 5), and the negative enrichment of CD40/CD40L signaling are all specific to the lung tissue of COVID-19 patients (S2 Fig in S1 Data). These results imply that the onset of the immune response in the lung tissue of COVID-19 patients may be correlated with elevated infiltration of naive B cells, which was further verified in single-cell RNA-seq databases (Figs 6 and 7).

The following hypothesis was developed based on the results of functional gene-set analysis combined with in silico immune infiltration profiling, which is similar to the context of influenza-specific B cell response [65]: SARS-CoV-2 infection may activate and promote the adhesion and accumulation of naïve B cells to the mediastinal lymph node through the activation of β2 integrin (LFA-1) and α4β1 integrin [57, 66], resulting in a decreased proportion of naïve B cells in the peripheral blood [67, 68]. The increased infiltration of naïve B cells activated by spike proteins of SARS-CoV-2 secretes large amounts of IgM to promote humoral immune response [65, 69]. Monocytes are recruited and differentiate to macrophage in response to the amplified humoral immune response [70]. Secreted IgM simultaneously activates the complement system and Fc receptor in dendritic cells and macrophage to increase antigen presentation and phagocytosis to facilitate innate and adaptive immunity [71, 72]. In addition, abundant spike protein from SARS-CoV-2 presented by antigen-presenting cell (APC) may leads to rapid induction of extrafollicular (EF) response through stimulating IL-12-dependent plasma cell differentiation in naïve B cell to produce more IgM [65, 73], instead of promoting B cell proliferation and differentiation by CD40/CD40L signaling mediated germinal center formation [74].

Based on this hypothesis, we believe that by suppressing the growth or migration of naïve B cells, it may help to reduce the excessive immune response caused by naïve B cells associated humoral immune response to reduce the risk of lung damage after SARS-CoV-2 infection, which have also been reported recently [75]. Remarkably, Quinti et al. reported several COVID-19 patients with primary antibody deficiencies (PAD) who clinically exhibited strikingly different extents of symptoms [76]. Five of the patients with Common Variable Iummune Deficiency (CVID) had severe COVID-19 symptoms. B cells in patients with CVID fail to differentiate into memory B cells, which maintain their properties similar to naïve B cells and continue to release IgM and IgG [77]. In contrast, 2 COVID-19 patients with agamma-globulinemia had mild symptoms and favorable outcome. These patients were congenitally deficient in B cells and plasma cells due to mutations in the gene encoding bruton kinase,

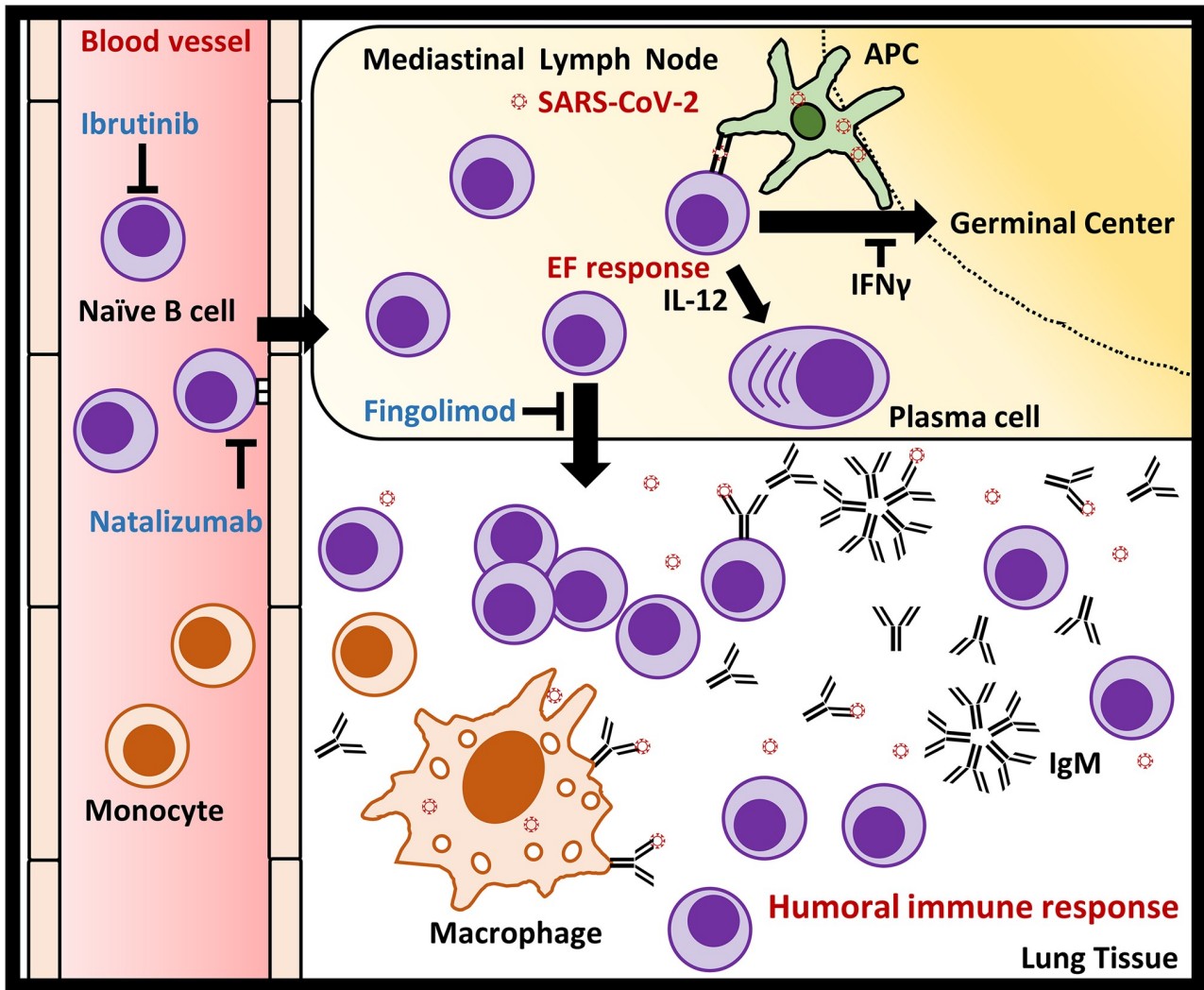

**Fig 8. Diagrammatic representation illustrate the postulated role of naïve B cell in triggering humoral immune response in lung tissues of COVID-19 patients and the locations where B cell targeted therapeutic strategies function.** SARS-CoV-2 infection may activate and promote the adhesion and accumulation of naïve B cells to the mediastinal lymph node with β2 integrin (LFA-1) and α4β1 integrin. The increased infiltration of naïve B cells activated by spike proteins from SARS-CoV-2 secretes large amounts of IgM to promote extrafollicular response and humoral immune response. Monocytes are recruited and differentiate to macrophage in response to the robust humoral immune response. Secreted IgM simultaneously activates the complement system and Fc receptor in dendritic cells and macrophage to increase antigen presentation and phagocytosis to facilitate innate and adaptive immunity. On the other hand, high affinity or abundant SARS-CoV-2 presented by antigen-presenting cell (APC) such as dendritic cell may stimulate IL-12-dependent plasma cell differentiation in naïve B cells to produce more IgM, instead of promoting B cell proliferation and differentiation by CD40/CD40L signaling mediated germinal center formation. Ibrutinib inhibits B-cell growth by specifically inhibiting Bruton kinase, which is thought to be critical for the BCR signaling pathway; Natalizumab reduces B-cell migration by blocking α4β1 integrin. Fingolimod reduces B-cell egress out of the lymph node by stimulating S1PR1/3 to internalize the receptors.

which is essential for B cell survival [78]. This report increases the likelihood of our hypothesis that naïve B cells act as the trigger of severe respiratory and pulmonary symptoms of COVID-19.

In the clinical treatment of abnormal increases in serum IgM or neoplastic B cells, ibrutinib, which inhibits the activity of Bruton Kinase in the B cell receptor signaling pathway [79], is now commonly used to reduce the abnormal proliferation of B cells [80, 81]. Treon et al.

reported that the use of ibrutinib may be useful to reduce lung damage from SARS-CoV-2 infection by suppressing the number of B cells [82]. Fingolimod, as an agonist of the S1P1/3 receptor, is thought to inhibit B cell egress out of lymph node through overstimulation of the B cell S1P1/3 receptor [83, 84]. Foerch et al. reported that multiple sclerosis patient with severe COVID-19 infection was being treated with Fingolimod. The patient improved rapidly right after appropriate therapy [85]. Integrin complexes have multiple implications in immune cell migration and viral infection. Immune cells use LFA-1 and α4β1 integrin for migration and adhesion [86, 87], which are also important for B cell [56, 57]. In addition, SARS-CoV-2 viral protein is known to bind to ACE2 or integrin heterodimers to facilitate virus entry and infection [88]. Borriello et al. reported on a COVID-19 patient who was using the α4β1 integrin targeted monoclonal antibody natalizumab [89], which functions to reduces B-cell migration by blocking α4β1 integrin and have significant effect in increasing circulating B cells [90, 91]. The patient improved significantly with appropriate treatment and no new symptoms developed or worsened. In terms of targeting interleukin, increased expression of IL-4 is thought to be associated with IPF, and dupilumab was also effective in asthma exacerbations, implying that inhibition of IL-4 may alleviate lung damage caused by SARS-CoV-2 by attenuating inflammation [92, 93]. Conversely, IL-12 is associated with the suppression of pulmonary fibrosis, and the clinical application in COVID-19 remains need to be further clarified [94, 95].

In summary, in silico simulated immune infiltration combined with gene function enrichment analysis provide us novel perspective on the immune system impact of SARS-CoV-2 infection. It is hoped that these findings will lead to new opportunities for the clinical treatment of COVID-19. All hypothetical mechanisms and locations of B cell targeted treatment acting are shown in Fig 8.

## Supporting information

**S1 Data.**
(PDF)

## Acknowledgments

We would like to thank Dr. Ting from Massachusetts General Hospital for uploading the valuable RNA-seq analysis results of organ samples from COVID-19 patients (NCBI GEO Access ID: GSE150316). We would also like to thank Dr. Kaminski at Yale University and Dr. Arron at Genentech, Inc. for uploading the valuable results of lung tissue sample analysis in IPF patients (NCBI GEO Access ID: GSE124685 and GSE53845).

## Limitations of the study

We understand that computer modelling of immuno-infiltration has its limits and problems of over-interpretation, and further experimental validation is often required. However, in the midst of the COVID-19 pandemic, we are eager to provide any significant data that can be used for interpretation or to increase confidence in COVID-19 treatment. Given the support of grant funding, the difficulty of collecting samples from COVID-19 patients, the limitations of the research environment and resources, and the fact that time is of the essence, we are strong-minded to announce the data as soon as possible, in the hope that we can do our part to contribute to the treatment of COVID-19.

## Author Contributions

**Conceptualization:** Sheng-Huei Wang, Ching-Liang Ho, Yi-Lin Chiu.

**Data curation:** Chih-Hsien Wu, Hsing-Fan Lai, Yi-Lin Chiu.

**Formal analysis:** Chih-Hsien Wu, Li-Chen Yen, Hsing-Fan Lai, Yi-Lin Chiu.

**Funding acquisition:** Yi-Ying Wu, Ching-Liang Ho, Yi-Lin Chiu.

**Investigation:** Sheng-Huei Wang.

**Project administration:** Yi-Lin Chiu.

**Supervision:** Ching-Liang Ho.

**Validation:** Chih-Hsien Wu, Li-Chen Yen, Hsing-Fan Lai.

**Visualization:** Li-Chen Yen.

**Writing – original draft:** Yi-Ying Wu, Sheng-Huei Wang, Yi-Lin Chiu.

**Writing – review & editing:** Yi-Ying Wu, Ching-Liang Ho, Yi-Lin Chiu.

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
