## [Decision Letter · Decision Letter 0]

7 Sep 2020

PONE-D-20-21528

In silico immune infiltration profiling combined with functional enrichment analysis reveals the specific role of naive B cells as a trigger for severe immune responses in the lungs of COVID-19 patients

PLOS ONE

Dear Dr. Chiu,

Thank you for submitting your manuscript to PLOS ONE. After careful consideration, we feel that it has merit but does not fully meet PLOS ONE’s publication criteria as it currently stands. Therefore, we invite you to submit a revised version of the manuscript that addresses the points raised during the review process.

The study is a very timely contribution by the authors to elucidate the etiology of lung damage in COVID-19 patients. In general, the manuscript is well written. The Reviewers and the Editor agree that this is an exciting study. We believe, addressing the Reviewer’s comments will significantly strengthen the manuscript.

We look forward to receiving your revised manuscript.

Kind regards,

Mrinmoy Sanyal, PhD

Academic Editor

PLOS ONE

Journal Requirements:

Reviewers' comments:

Reviewer's Responses to Questions

**Comments to the Author**

1. Is the manuscript technically sound, and do the data support the conclusions?

Reviewer #1: Partly

Reviewer #2: Partly

2. Has the statistical analysis been performed appropriately and rigorously? 

Reviewer #1: No

Reviewer #2: Yes

3. Have the authors made all data underlying the findings in their manuscript fully available?

Reviewer #1: Yes

Reviewer #2: Yes

4. Is the manuscript presented in an intelligible fashion and written in standard English?

Reviewer #1: Yes

Reviewer #2: Yes

5. Review Comments to the Author

Reviewer #1: Wu et al. employed CIBERSORT method to investigate common and differences in the level of immune cell infiltration in lung tissues of COVID-19 patients compared with patients of Idiopathic Pulmonary Fibrosis (IPF). They determined that several immune cell sub-types, particularly naive B-cells, are highly infiltrated in COVID-19 patients. Besides that they reported several other important findings using CIBERSORT and gene set enrichment based analysis (Figs 2-6). Overall this is an interesting study. However, the study is suffering from lack of validation results. Authors have stated this limitation and addressed this major issue only by indicating that in the midst of COVID-19 pandemic, they are eager to provide this data that can be used for interpretation or to increase confidence in COVID-19 treatment. Although rapid data generation is a critical step during this pandemic situation; however, ideally we should not compromise with reproducibility, scientific rigor, and quality of the data.

Major comments

1. It is understandable that the experimental validation of the major prediction results reported in this manuscript is a time consuming task; however, some levels of validation are required to determine the quality of the reported prediction results. Authors are suggested to evaluate single cell RNA-seq data and investigate whether the prediction results they obtained from the bulk RNA-seq data using the tool CIBERSORT are reproducible in the actual single cell resolution. A quick pubmed search shows, recently Liao et al. (Nature Medicine, May 2020; https://doi.org/10.1038/s41591-020-0901-9) published single cell landscape of bronchoalveolar immune cells extracted from moderate to sever patients with COVID-19 and healthy control and the data are publicly available (GSE128033 and GSM3660650). Authors are suggested to do a comprehensive literature search and check if this study or any additional studies could be used as the validation data. Furthermore, multiple single cell RNA-seq data on IPF (GSE94555 and GSE86618) are also available in the public repositories. Therefore, authors are suggested to take this opportunity and investigate whether their reported comparative results are reproducible in a completely independent validation data set.

2. If the single cell data are not suitable for the analysis because of some valid reasons, authors are suggested to at least evaluate independent RNA-seq data and examine whether the prediction results obtained from the training data are consistent in the independent validation data.

3. For the Gene set enrichment analysis FDR cut-off 0.25 was chosen to determine significantly enriched gene sets. Although this cut-off is not uncommon; however, a general trend is to use FDR < 0.05 to determine significance. I am wondering how much the results would be changed with the FDR < 0.05 compared with the current results. This is important to evaluate, particularly in the absence of validation data, because such a low cut-off value may introduce high-false positives.

Reviewer #2: Summary:

This is a timely manuscript presenting a comparative in silico analysis of estimated cell type frequencies and gene expression changes in the lungs of deceased COVID-19 patients and idiopathic pulmonary fibrosis patients. Using CIBERSORT to estimate cell subset abundancies from gene expression profiles, the authors find certain cell types, such as monocytes and naïve B cells, uniquely increased in fatal COVID-19 samples. Furthermore, gene set enrichment analysis revealed changes in functional and signaling pathways, including decreases in CD40/CD40L signaling and alterations in integrin expression. The authors suggest that these results implicate naïve B cells as a potential mediator of lung pathology in COVID-19. Overall, this manuscript presents original research and the conclusions are supported by the data. However, there are some concerns about the limitations of the study and presentation of the results that should be addressed prior to acceptance.

Major concerns:

1. The title overstates the findings and should be changed to reflect the fact that the role for naïve B cells in COVID-19 lung pathology is not proven by this analysis. A potential title could be: “In silico immune infiltration profiling combined with functional enrichment analysis [suggests a/reveals a potential] role for naïve B cells as a trigger for severe immune responses in the lungs of COVID-19 patients”.

2. A significant limitation which is not addressed is that all of the COVID-19 samples analyzed were from fatal cases. To understand whether naïve B cell infiltration plays a specific role in the pathology of severe/fatal COVID-19, it would be important to examine whether these changes also occur in mild/moderate cases where there is a lack of significant lung damage or fibrosis. If the samples/data are not available to make this comparison, this limitation should be raised in the discussion.

3. The authors state the following rationale for comparing COVID-19 and IPF: “Nearly all of the patients who died from COVID-19 had severe lung tissue damage and pulmonary fibrosis[29]. On the other hand, mortality in IPF is generally the result of progressive fibrotic lung disease.” If pulmonary fibrosis is a cause of mortality in both cases, it would make sense to look for common signatures rather than those unique to COVID-19. I think further clarification on the rationale here would be useful.

4. Figures 2-4: CIBERSORT can return relative cellular fractions instead of absolute score. These would be easier to interpret as an estimated percentage of total cells. In some cases the y axis scale changes significantly between datasets, making it harder to compare. Additionally, as a positive control it would be good to show that the relative cellular fractions of different subsets in the healthy samples match the approximate expected distribution of cell types in normal lung tissue. This could be included as a supplementary figure.

5. Figure 5: Given that there are ~7000 GO gene sets, an FDR<0.25 cutoff is quite relaxed and will likely result in many false positives. Do the results change if using a stricter cutoff such as 0.05?

6. Figure 5-6: In panel 5A, it appears the results from the two IPF datasets have been merged, but in panel 5B and Figure 6 they are separated. The authors should merge these results as well in panel 5B/Figure 6 to simplify the visualization and make a clearer message. Furthermore, the method of merging is not described. I would suggest running the enrichment analysis separately in each dataset and identifying gene sets that are consistently enriched in both datasets to produce the most robust results.

7. Figure 5B: The figure legend mentions “positive correlations”. If these are correlations, what are the variables being correlated? Instead, are these gene set enrichments? If so, the authors should change “positive correlations” to “positive enrichments” or “upregulated pathways versus healthy”. The legend also states that the edge represent overlap in genes between different gene sets. If this is so, how can an edge be dataset specific (colored), since the members of a gene set are predefined and not dataset dependent?

8. Figures 5/6: As is suggested by the differences in estimated cell frequencies in Figures 2-4, the changes in gene expression between COVID-19/IPF and healthy can be driven both by alterations in cell type abundancies and by transcriptional changes within a given cell type. The authors of CIBERSORT recently released a newer version, CIBERSORTx (Newman et al. Nat. Biotech. 2019) which allows estimation of cell type-specific gene expression within a mixture. I believe a valuable extension to the analysis presented would be to compare estimated gene expression between COVID-19/IPF and healthy within particular subsets of interest, such as naïve/memory B cells. This could provide further clarity, for example by revealing whether the upregulation of integrin pathways is occurring within naïve B cells of COVID-19 patients or if it is occurring in other cell types or simply due to changes in B cell frequencies.

9. Figure 6: The results section describing changes in various interleukin signaling pathways is not clear about the disease specificity. It should mention that these changes are common to IPF and COVID-19. Furthermore, IL-4 signaling, which is upregulated in COVID-19 (and IPF), is known to induce proliferation and class switching in B cells (Gascan et al., J. Exp. Med. 1991). This appears at odds with the finding of increased naïve B cells in COVID-19 samples. How do the authors explain this apparent dichotomy? This should be addressed.

Minor concerns:

1. Figure 5B: What does the size of the circles represent?

2. Figure 6: Are these enrichments significant? The cutoffs used to determine inclusion in the figure should be stated in the legend

3. Discussion: The authors refer to increased infiltration of different immune cells, it should be clarified that these are transcriptional estimates and not actual measurements of cell frequencies.

6. PLOS authors have the option to publish the peer review history of their article (what does this mean?). If published, this will include your full peer review and any attached files.

Reviewer #1: No

Reviewer #2: No

---

## [Author Response · Author response to Decision Letter 0]

20 Oct 2020

Response to reviewer 1:

Dear reviewer, thank you very much for your valuable comments and suggestions on the direction of this paper, which have greatly help us improving the strength of evidence of our conclusions. Based on the suggestion, we attempted to search multiple public repositories and found the single-cell sequencing databases currently associated with COVID-19 lung-related immune cell infiltration provided by Chua et al. and Liao et al. (GSE145926), and the single-cell sequencing database associated with IPF lung immune cell infiltration published by Morse et al. (GSE128033). The GSE94555 and GSE86618 databases were also once included in our evaluation but were not used due to the small sample size. On the other hand, the GSE135893 IPF database contains a large number of single-cell analysis results, but due to the processing capacity of the equipment, it was not possible to analyze the relevant results of the B-cell population. 

In order to achieve consistency with the CIBERSORT analysis to verify whether similar results occur in both the bulk RNA-seq and single-cell RNA-seq, and to allow for re-running the gene set clustering and visualization analysis with FDR < 0.05, we performed gene cluster overlap visualization analysis by matching the immune cell signature files used in CIBERSORT and converting them to Naïve or Memory specific gene sets. In addition, in order to reduce the complexity and improve the interpretation of the results, we polarized the enrichment results as red and green for COVID-19 or IPF respectively. The results of the single cell sequencing database and clinical situation analysis were added to Figure 6 and 7, while the PID-related analysis was included as supplementary data. For B-cell related signature analysis, since the number of gene sets has been filtered down to nearly 500, the use of FDR<0.05 may result in too few results, so please forgive us for adopting the more relaxed FDR<0.25 criterion in Figure 7A. The relevant amendments and new result paragraphs are described in the attached rebuttal letter.

Responses to Reviewer 2:

1. The title overstates the findings and should be changed to reflect the fact that the role for naïve B cells in COVID-19 lung pathology is not proven by this analysis. A potential title could be: “In silico immune infiltration profiling combined with functional enrichment analysis [suggests a/reveals a potential] role for naïve B cells as a trigger for severe immune responses in the lungs of COVID-19 patients”.

Response:

Many thanks to the valuable suggestion, we have corrected the title as follows: “In silico immune infiltration profiling combined with functional enrichment analysis reveals a potential role for naïve B cells as a trigger for severe immune responses in the lungs of COVID-19 patients.”

2. A significant limitation which is not addressed is that all of the COVID-19 samples analyzed were from fatal cases. To understand whether naïve B cell infiltration plays a specific role in the pathology of severe/fatal COVID-19, it would be important to examine whether these changes also occur in mild/moderate cases where there is a lack of significant lung damage or fibrosis. If the samples/data are not available to make this comparison, this limitation should be raised in the discussion.

Response: 

We thank the reviewer for the suggestion. For moderate/severe case comparison, we analyzed two single cell RNA-seq databases that clearly indicate the severity of COVID-19, and further compared the phenotypic differences between B cells from different conditions and their degree of infiltration in relation to clinical status in Figure 7. The relevant amendments are described in the rebuttal letter.

3. The authors state the following rationale for comparing COVID-19 and IPF: “Nearly all of the patients who died from COVID-19 had severe lung tissue damage and pulmonary fibrosis[29]. On the other hand, mortality in IPF is generally the result of progressive fibrotic lung disease.” If pulmonary fibrosis is a cause of mortality in both cases, it would make sense to look for common signatures rather than those unique to COVID-19. I think further clarification on the rationale here would be useful.

Response: 

We thank the reviewer for the suggestions and strongly agree that it is necessary and valuable to analyze the shared signatures of the two diseases. However, given the availability of raw data and the complex interactions of multiple immune cells, we are currently unable to analyze such a large amount of data and further expand the presentation of the results. For example, the data tentatively suggest that differences in macrophage are common to both diseases, a phenomenon also observed in our newly added single cell sequencing database, but a complete analysis of all macrophage-related gene set signatures and correlations would require more time, knowledge, and manpower to achieve. On the other hand, we did observe some possible disease-related findings, such as the suppression of insulin-associated signaling pathways in both diseases, which we believe may be related to the abnormal regulation of blood glucose in patients with pulmonary fibrosis, but decided to delete this discussion due to space limitation and lack of expertise. As such, we are grateful for your understanding of our decision to focus on the specific differences between the two diseases. Hopefully, the publication of this paper will allow more experts in various fields to conduct in-depth analysis on different faces.

4. Figures 2-4: CIBERSORT can return relative cellular fractions instead of absolute score. These would be easier to interpret as an estimated percentage of total cells. In some cases the y axis scale changes significantly between datasets, making it harder to compare. Additionally, as a positive control it would be good to show that the relative cellular fractions of different subsets in the healthy samples match the approximate expected distribution of cell types in normal lung tissue. This could be included as a supplementary figure.

Response：

Thank you very much for your suggestion and we couldn't agree with you more. We have evaluated the relative scores using Cibersort-X, but the presentation of the results may be affected by the over-representation of specific immune cells, which significantly reduces the impact of the analysis of the target cell clusters. On the other hand, there are other recent publications that analyzed the same database using different deconvolution strategies, and we decided not to include the presentation and discussion of relative cell scores in this paper due to the issue of repetitive publication.

Reference:

Cavalli, E., et al., Transcriptomic analysis of COVID19 lungs and bronchoalveolar lavage fluid samples reveals predominant B cell activation responses to infection. Int J Mol Med, 2020. 46(4): p. 1266-1273.

5. Figure 5: Given that there are ~7000 GO gene sets, an FDR<0.25 cutoff is quite relaxed and will likely result in many false positives. Do the results change if using a stricter cutoff such as 0.05?

Response：

Thanks to the reviewer's suggestion, we have recreated Figure 5 under stricter conditions, as described in the rebuttal letter.

6. Figure 5-6: In panel 5A, it appears the results from the two IPF datasets have been merged, but in panel 5B and Figure 6 they are separated. The authors should merge these results as well in panel 5B/Figure 6 to simplify the visualization and make a clearer message. Furthermore, the method of merging is not described. I would suggest running the enrichment analysis separately in each dataset and identifying gene sets that are consistently enriched in both datasets to produce the most robust results.

Response：

Thanks to the reviewer's suggestion, we have recreated Figure 5, which combines the IPF database results and makes the visualization easier to interpret. All corrections and results will be described below.

7. Figure 5B: The figure legend mentions “positive correlations”. If these are correlations, what are the variables being correlated? Instead, are these gene set enrichments? If so, the authors should change “positive correlations” to “positive enrichments” or “upregulated pathways versus healthy”. The legend also states that the edge represent overlap in genes between different gene sets. If this is so, how can an edge be dataset specific (colored), since the members of a gene set are predefined and not dataset dependent?

Response：

Thanks to the reviewer for the correction, this is indeed an error in our narration. On the one hand, the phenomenon of edge appearing colored no longer appears after updating the enrichment map, which may be a version-specific anomaly that causes us to misinterpret the results. The above error has been corrected after reproduction and is described below.

8. Figures 5/6: As is suggested by the differences in estimated cell frequencies in Figures 2-4, the changes in gene expression between COVID-19/IPF and healthy can be driven both by alterations in cell type abundancies and by transcriptional changes within a given cell type. The authors of CIBERSORT recently released a newer version, CIBERSORTx (Newman et al. Nat. Biotech. 2019) which allows estimation of cell type-specific gene expression within a mixture. I believe a valuable extension to the analysis presented would be to compare estimated gene expression between COVID-19/IPF and healthy within particular subsets of interest, such as naïve/memory B cells. This could provide further clarity, for example by revealing whether the upregulation of integrin pathways is occurring within naïve B cells of COVID-19 patients or if it is occurring in other cell types or simply due to changes in B cell frequencies.

Response:

We would like to thank Reviewer for his suggestion to re-analyze the results in Cibersort-X, but the data is too complex. For example, the specific genes screened by the software are currently unsupported by literature and are difficult to link to Naïve or memory B cells. Therefore, we used the LM22 immune cell gene signature from Cibersort to create specific signatures for Naïve or memory B cell, and then analyzed the bulk RNA-seq and single-cell RNA-seq to cross-validate the enrichment of the same signature in a wider variety of databases. Related gene expression is also specifically evaluated. We believe that such an approach would enhance the credibility of the results as well. All corrections and results have be described in the rebuttal letter.

9. Figure 6: The results section describing changes in various interleukin signaling pathways is not clear about the disease specificity. It should mention that these changes are common to IPF and COVID-19. Furthermore, IL-4 signaling, which is upregulated in COVID-19 (and IPF), is known to induce proliferation and class switching in B cells (Gascan et al., J. Exp. Med. 1991). This appears at odds with the finding of increased naïve B cells in COVID-19 samples. How do the authors explain this apparent dichotomy? This should be addressed.

Response:

Thanks to Reviewer's suggestion, we understand that it is difficult to directly relate the results of bulk RNA-seq samples to the effects of specific cell populations. However, it is possible that presenting the results of this kind of analysis may provide preliminary confirmation for other more in-depth studies, so we decided to present the results in supplementary data. On the other hand, we also tried to perform PID analysis in single cell RNA-seq, but the single B-cell diversity may be too divergent for simple dichotomy to obtain significant results, and thus the results are not presented. On the issue of IL-4 signaling, a report published this year indicated that the activation of IL-4 signaling may be related to the restore of DN2 B cell to naïve B cell. Although this result is somewhat contrary to our additional inference, since the role of DN2 B cells in autoimmunity and the detailed activation mechanism remain to be elucidated, the up-regulation of IL-4 signal may also be a potential inhibitory mechanism indirectly induced by the over-activation of DN2 B cells, and preserving the relevant data may help future research progress.

Reference:

Hsu, H.-C., et al., IL-4 synergizes with low-dose IL-2 to restore systemic lupus erythematosus B cells at the resting naive status. The Journal of Immunology, 2020. 204(1 Supplement): p. 218.8-218.8.

Minor concerns:

1. Figure 5B: What does the size of the circles represent?

Response:

Thanks to reviewer's correction, the size of the circle, which represents the number of genes in the gene set, is clearly indicated in the recreated figure.

2. Figure 6: Are these enrichments significant? The cutoffs used to determine inclusion in the figure should be stated in the legend

Response:

Thanks to reviewer's correction, all enrichments have been filtered for FDR < 0.05 and are noted specifically in the recreated diagram and legends.

3. Discussion: The authors refer to increased infiltration of different immune cells, it should be clarified that these are transcriptional estimates and not actual measurements of cell frequencies.

Response:

Thanks to the reviewer for the corrections, which have been made to the text.

---

## [Decision Letter · Decision Letter 1]

6 Nov 2020

PONE-D-20-21528R1

In silico immune infiltration profiling combined with functional enrichment analysis reveals a potential role for naive B cells as a trigger for severe immune responses in the lungs of COVID-19 patients

PLOS ONE

Dear Dr. Chiu,

Thank you for submitting your manuscript to PLOS ONE. After careful consideration, we feel that it has merit but does not fully meet PLOS ONE’s publication criteria as it currently stands. Therefore, we invite you to submit a revised version of the manuscript that addresses the points raised during the review process.

We look forward to receiving your revised manuscript.

Kind regards,

Mrinmoy Sanyal, PhD

Academic Editor

PLOS ONE

Reviewers' comments:

Reviewer's Responses to Questions

**Comments to the Author**

1. If the authors have adequately addressed your comments raised in a previous round of review and you feel that this manuscript is now acceptable for publication, you may indicate that here to bypass the “Comments to the Author” section, enter your conflict of interest statement in the “Confidential to Editor” section, and submit your "Accept" recommendation.

Reviewer #1: All comments have been addressed

Reviewer #2: (No Response)

2. Is the manuscript technically sound, and do the data support the conclusions?

Reviewer #1: (No Response)

Reviewer #2: Partly

3. Has the statistical analysis been performed appropriately and rigorously? 

Reviewer #1: (No Response)

Reviewer #2: Yes

4. Have the authors made all data underlying the findings in their manuscript fully available?

Reviewer #1: (No Response)

Reviewer #2: Yes

5. Is the manuscript presented in an intelligible fashion and written in standard English?

Reviewer #1: (No Response)

Reviewer #2: Yes

6. Review Comments to the Author

Reviewer #1: (No Response)

Reviewer #2: Summary:

The authors have addressed most of my initial concerns. However, the updated analysis and results suggest that naïve B cells are not elevated in the BALF of severe COVID-19 patients relative to moderate cases, but that the B cells of severe patients may be enriched in antibody-secreting cells instead. This does not implicate naïve B cells in severe immune responses to COVID-19 and the conclusions of the manuscript need to be altered to be in agreement with these new findings.

Major concerns:

1. Figure 5B: How do the authors define ‘top gene’ in the memory v naïve comparison? Is this based on the top 100 most highly expressed genes from the LM22 signature for both the memory and naïve B cell subsets? This is not an ideal way to define a naïve vs memory specific signature, because it is not guaranteed to identify the most distinctive genes. The 20 genes unique to naïve could be rank 101-120 in memory, just missing the cutoff, for example. Therefore, the authors should validate their analyses by using an alternative memory vs naïve signature and see if they produce consistent results. One approach could be to use up/down DEG memory vs naïve B cell gene sets included in the C7:immunologic signature gene sets from MSigDB.

2. Figure 7: The the moderate versus severe B cell comparison here is interesting and informative. However, as the authors correctly point out, the results indicate that severe patients likely have increased plasma cells or other antibody-secreting cells relative to mild/moderate patients. These cells, as well as DN2 B cells, are not naïve. Therefore, this would suggest that naïve B cells are not involved in the pathology of severe COVID-19, but that potentially antibody-secreting cells are instead. This is at odds with the conclusions of the manuscript, including the title. Unless the authors have some alternative explanation as to why these results do not implicate antibody-secreting cells rather than naïve B cells in severe COVID-19, the conclusions of the manuscript need to be changed to better match these findings.

Minor concerns:

1. The end of the introduction mentions ‘anti-secretory cells’, is this supposed to be antibody-secreting cells (as in the abstract)?

2. Figure 5A: This panel might be easier to interpret if these results were separated into 3 plots (COVID-19, IPF, and common)

3. Figure 6: The Liao and Morse datasets both have ~40% B cells in control samples, but the Chua dataset has < 1% B cells in control samples. Why do these controls have such a low frequency of B cells? Are they appropriate controls?

4. Figure 7B: What is the source for these DN2 B cell gene markers? Are these differences significant?

7. PLOS authors have the option to publish the peer review history of their article (what does this mean?). If published, this will include your full peer review and any attached files.

Reviewer #1: No

Reviewer #2: No

---

## [Author Response · Author response to Decision Letter 1]

10 Nov 2020

Reviewer #2: Summary:

The authors have addressed most of my initial concerns. However, the updated analysis and results suggest that naïve B cells are not elevated in the BALF of severe COVID-19 patients relative to moderate cases, but that the B cells of severe patients may be enriched in antibody-secreting cells instead. This does not implicate naïve B cells in severe immune responses to COVID-19 and the conclusions of the manuscript need to be altered to be in agreement with these new findings.

Major concerns:

1. Figure 5B: How do the authors define ‘top gene’ in the memory v naïve comparison? Is this based on the top 100 most highly expressed genes from the LM22 signature for both the memory and naïve B cell subsets? This is not an ideal way to define a naïve vs memory specific signature, because it is not guaranteed to identify the most distinctive genes. The 20 genes unique to naïve could be rank 101-120 in memory, just missing the cutoff, for example. Therefore, the authors should validate their analyses by using an alternative memory vs naïve signature and see if they produce consistent results. One approach could be to use up/down DEG memory vs naïve B cell gene sets included in the C7:immunologic signature gene sets from MSigDB.

Response:

We really appreciate the suggestions, and we have indeed tried and considered a number of methods similar to this. We initially spent a lot of time searching the literature for a more representative gene-set of memory and naïve B-cell, but most of the literature only had one or two genes (e.g. CD27 for memory B cell) as representative of the B-cell population. Considering that we initially observed this phenomenon from the results of the Cibersort analysis and aimed to narrow down the results to focus on differences in expression of immune cell related genes in further single-cell sequencing databases, we finally decided to use the current method for the evaluation. There are two reasons for choosing the top 100 genes: firstly, in the LM22 document, the weighting (numbers of each gene in the LM22 document ) of the top 10 immune cell-specific genes is over 10,000 and drops to a few hundred after the top 100, and the weighting of the genes further down the list is insignificant for the software to determine the immune infiltration score of the immune cells. Secondly, among the immune cell-specific genes generated by this method, although there are indeed problems as you mentioned, most of them have more than twofold differences in the weight of genes in the LM22 naive and memory cell gene set, so we think that they still efficient to discriminate between the two types of B cells. Further, a number of genes are listed as naive or memory B-cell-specific biomarkers in the single-cell biomarker database (PanglaoDB, https://www.panglaodb.se/index.html) (Naive: ZNF286A, MEP1A, FCER2, UGT1A8, BEND5, BCL7A, P2RY14; Memory: TNFRSF13B, CD27, CD86, IL7, AIM2) , and most of the biomarker genes fall in the 50th to 100th rank, showing that this method is capable of accurately reflecting biomarker genes in specific immune cell type. In addition, we also used the same method to find specific genes representing memory and naive B cells in the MSigDB C7:Immunology signature gene set (GSE13411), with even fewer genes intersecting with those from single-cell biomarker database, and thus finally decided not to adopt it for analysis.

2. Figure 7: The moderate versus severe B cell comparison here is interesting and informative. However, as the authors correctly point out, the results indicate that severe patients likely have increased plasma cells or other antibody-secreting cells relative to mild/moderate patients. These cells, as well as DN2 B cells, are not naïve. Therefore, this would suggest that naïve B cells are not involved in the pathology of severe COVID-19, but that potentially antibody-secreting cells are instead. This is at odds with the conclusions of the manuscript, including the title. Unless the authors have some alternative explanation as to why these results do not implicate antibody-secreting cells rather than naïve B cells in severe COVID-19, the conclusions of the manuscript need to be changed to better match these findings.

Response：Thank you for your approval. With regard to the association of antibody-secreting cells with naive cells, we did not adequately describe the context in the current manuscript and have therefore corrected it as shown below:

“As possessing both naïve B-cell and antibody-secreting cell properties, these cells are similar to the DN2 B-cells found in systemic lupus erythematosus proposed by Tipton et al. The researchers found that they are derived from naïve B cells and appear to be precursors of plasma cells [1]. The term double negative (DN) comes from the absence of immunoglobulin D and memory B-cell marker CD27 [2]. DN B cells are further classified into DN1(CXCR5+) and DN2(CXCR5-) by whether CXCR5 is expressed or not [3]. An increase in IRF4 and a decrease in IRF8 are thought to be associated with the promote naïve B cell differentiation into DN2 [4, 5]. The process of induced DN2 B-cell differentiation is accompanied by a decrease in TRAF5 and CD21 and an increase in CD11c [6]. These DN2 B cells are usually found outsides of B-cell follicles rather than germinal centers and may therefore be genetically phenotypically different from GC-derived memory B cells.”

Regarding the difference between antibody-secreting cells (ASC) and naive cells, it is true that it is difficult to connect the two in previous definitions. However, recent studies investigating DN2 B cells that induce Systemic Lupus Erythematosus (SLE) indicate that these B cells are directly stimulated by follicular helper T cells and differentiate into plasma-like cells that secrete large amounts of antibodies while retaining properties of naïve cells to some extent [6]. These implications led us to try to assess whether the infiltrating B-cells in patients with severe COVID-19 have a tendency to be DN2 B-cells.

We understand that there is insufficient evidence to directly define these ASC-like naïve cells as DN2 B cells by inference for the further conclusion in this article, because there is no credible gene set or experiment to define these naïve B cells as DN2 B cells so far. Therefore, we decided to keep the initial “naive B-cells” in the title, which showed a similar tendency in both bulk RNA-seq and single cell sequencing results and included the postulated characteristics of DN2 B cell in the single cell sequencing analysis and discussion as a clue for further research.

Minor concerns:

1. The end of the introduction mentions ‘anti-secretory cells’, is this supposed to be antibody-secreting cells (as in the abstract)?

Response：

Thank you for correcting this typo, which we have corrected as follows:

“Further analysis of the defined B-cell population using single-cell RNA sequencing databases showed that the B-cells from COVID-19 patients not only tended to be naïve B-cells, but also tended to be antibody-secreting cells in patients with severe disease, and the proportion of B-cell infiltration seemed to correlate with the severity of the disease.”

2. Figure 5A: This panel might be easier to interpret if these results were separated into 3 plots (COVID-19, IPF, and common)

Response：

Thank you for your suggestion, we have recreated the following figure: 

3. Figure 6: The Liao and Morse datasets both have ~40% B cells in control samples, but the Chua dataset has < 1% B cells in control samples. Why do these controls have such a low frequency of B cells? Are they appropriate controls?

Response：

With regard to the differences in B-cell infiltration in healthy donors, we suggest that this may be due to sampling methods. In the Liao and Morse datasets, samples were collected from patients' bronchoalveolar lavage fluid, whereas in Chua dataset, most samples were collected from respiratory mucosa by swab only. Therefore, in the Chua dataset, samples from healthy donors may have significantly different B-cell population due to lower mucosal secretions. We understand that this is an experimental limitation and that extreme differences in quantity may also cause bias in the results. However, this database is an important source of evidence for the extent of disease and immune cell infiltration in patients, and we felt it was necessary to perform consistent analyses across all databases, and therefore used these small numbers of B cells as controls.

4. Figure 7B: What is the source for these DN2 B cell gene markers? Are these differences significant?

Response:

Thank you for the consideration, these markers are based on the gene markers used in several widely cited papers discussing DN2 B-cells, so we trust that these markers are approved. The relevant genes and literature have been added to the context as follows:

“As possessing both naïve B-cell and antibody-secreting cell properties, these cells are similar to the DN2 (Double negative) B-cells found in systemic lupus erythematosus proposed by Tipton et al. The researchers found that they are derived from naïve B cells and appear to be precursors of plasma cells [1]. The term double negative (DN) comes from the absence of immunoglobulin D and memory B-cell marker CD27 [2]. DN B cells are further classified into DN1(CXCR5+) and DN2(CXCR5-) by whether CXCR5 is expressed or not [3]. An increase in IRF4 and a decrease in IRF8 are thought to be associated with the promote naïve B cell differentiation into DN2 [4, 5]. The process of induced DN2 B-cell differentiation is accompanied by a decrease in TRAF5 and CD21 and an increase in CD11c [6]. These DN2 B cells are usually found outsides of B-cell follicles rather than germinal centers and may therefore be genetically phenotypically different from GC-derived B cells.”

The relevant statistical difference calculations have been supplemented as shown in the figure below:

 

1. Tipton, C.M., et al., Diversity, cellular origin and autoreactivity of antibody-secreting cell population expansions in acute systemic lupus erythematosus. Nat Immunol, 2015. 16(7): p. 755-65.

2. Wei, C., et al., A new population of cells lacking expression of CD27 represents a notable component of the B cell memory compartment in systemic lupus erythematosus. J Immunol, 2007. 178(10): p. 6624-33.

3. Ehrhardt, G.R., et al., Discriminating gene expression profiles of memory B cell subpopulations. J Exp Med, 2008. 205(8): p. 1807-17.

4. Nutt, S.L., et al., The generation of antibody-secreting plasma cells. Nat Rev Immunol, 2015. 15(3): p. 160-71.

5. Xu, H., et al., Regulation of bifurcating B cell trajectories by mutual antagonism between transcription factors IRF4 and IRF8. Nat Immunol, 2015. 16(12): p. 1274-81.

6. Jenks, S.A., et al., Distinct Effector B Cells Induced by Unregulated Toll-like Receptor 7 Contribute to Pathogenic Responses in Systemic Lupus Erythematosus. Immunity, 2018. 49(4): p. 725-739 e6.

---

## [Decision Letter · Decision Letter 2]

12 Nov 2020

In silico immune infiltration profiling combined with functional enrichment analysis reveals a potential role for naive B cells as a trigger for severe immune responses in the lungs of COVID-19 patients

PONE-D-20-21528R2

Dear Dr. Chiu,

We’re pleased to inform you that your manuscript has been judged scientifically suitable for publication and will be formally accepted for publication once it meets all outstanding technical requirements.

Kind regards,

Mrinmoy Sanyal, PhD

Academic Editor

PLOS ONE

Reviewers' comments:

Reviewer's Responses to Questions

**Comments to the Author**

1. If the authors have adequately addressed your comments raised in a previous round of review and you feel that this manuscript is now acceptable for publication, you may indicate that here to bypass the “Comments to the Author” section, enter your conflict of interest statement in the “Confidential to Editor” section, and submit your "Accept" recommendation.

Reviewer #2: All comments have been addressed

2. Is the manuscript technically sound, and do the data support the conclusions?

Reviewer #2: (No Response)

3. Has the statistical analysis been performed appropriately and rigorously? 

Reviewer #2: (No Response)

4. Have the authors made all data underlying the findings in their manuscript fully available?

Reviewer #2: (No Response)

5. Is the manuscript presented in an intelligible fashion and written in standard English?

Reviewer #2: (No Response)

6. Review Comments to the Author

Reviewer #2: (No Response)

7. PLOS authors have the option to publish the peer review history of their article (what does this mean?). If published, this will include your full peer review and any attached files.

Reviewer #2: No

---

## [Editor Report · Acceptance letter]

17 Nov 2020

PONE-D-20-21528R2 

In silico immune infiltration profiling combined with functional enrichment analysis reveals a potential role for naïve B cells as a trigger for severe immune responses in the lungs of COVID-19 patients 

Dear Dr. Chiu:

I'm pleased to inform you that your manuscript has been deemed suitable for publication in PLOS ONE. Congratulations! Your manuscript is now with our production department. 

Kind regards, 

on behalf of

Dr. Mrinmoy Sanyal 

Academic Editor

PLOS ONE